# A Novel CXCR4-Targeted Diphtheria Toxin Nanoparticle Inhibits Invasion and Metastatic Dissemination in a Head and Neck Squamous Cell Carcinoma Mouse Model

**DOI:** 10.3390/pharmaceutics14040887

**Published:** 2022-04-18

**Authors:** Elisa Rioja-Blanco, Alberto Gallardo, Irene Arroyo-Solera, Patricia Álamo, Isolda Casanova, Ugutz Unzueta, Naroa Serna, Laura Sánchez-García, Miquel Quer, Antonio Villaverde, Esther Vázquez, Xavier León, Lorena Alba-Castellón, Ramon Mangues

**Affiliations:** 1Institut d’Investigació Biomèdica Sant Pau (IIB-Sant Pau), Sant Quintí, 77, 08041 Barcelona, Spain; erioja@santpau.cat (E.R.-B.); agallardoa@santpau.cat (A.G.); iarroyo@santpau.cat (I.A.-S.); palamo@santpau.cat (P.Á.); icasanova@santpau.cat (I.C.); uunzueta@santpau.cat (U.U.); 2Institut de Recerca contra la Leucèmia Josep Carreras, 08025 Barcelona, Spain; 3Department of Pathology, Hospital de la Santa Creu i Sant Pau, Sant Quintí, 89, 08041 Barcelona, Spain; 4CIBER de Bioingeniería, Biomateriales y Nanomedicina (CIBER-BBN), Monforte de Lemos 3-5, 28029 Madrid, Spain; naroas@nanoligent.com (N.S.); laurasanchezgarcia92@gmail.com (L.S.-G.); mquer@santpau.cat (M.Q.); antoni.villaverde@uab.cat (A.V.); xleon@santpau.cat (X.L.); 5Departament de Genètica i de Microbiologia, Universitat Autònoma de Barcelona, 08193 Bellaterra, Spain; 6Institut de Biotecnologia i de Biomedicina, Universitat Autònoma de Barcelona, 08193 Bellaterra, Spain; 7Department of Otorhinolaryngology, Hospital de la Santa Creu i Sant Pau, Universitat Autònoma de Barcelona, Sant Quintí, 89, 08041 Barcelona, Spain

**Keywords:** CXCR4, head and neck cancer, targeted drug delivery, protein nanoparticles, diphtheria toxin, metastasis

## Abstract

Loco-regional recurrences and metastasis represent the leading causes of death in head and neck squamous cell carcinoma (HNSCC) patients, highlighting the need for novel therapies. Chemokine receptor 4 (CXCR4) has been related to loco-regional and distant recurrence and worse patient prognosis. In this regard, we developed a novel protein nanoparticle, T22-DITOX-H6, aiming to selectively deliver the diphtheria toxin cytotoxic domain to CXCR4^+^ HNSCC cells. The antimetastatic effect of T22-DITOX-H6 was evaluated in vivo in an orthotopic mouse model. IVIS imaging system was utilized to assess the metastatic dissemination in the mouse model. Immunohistochemistry and histopathological analyses were used to study the CXCR4 expression in the cancer cells, to evaluate the effect of the nanotoxin treatment, and its potential off-target toxicity. In this study, we report that CXCR4^+^ cancer cells were present in the invasive tumor front in an orthotopic mouse model. Upon repeated T22-DITOX-H6 administration, the number of CXCR4^+^ cancer cells was significantly reduced. Similarly, nanotoxin treatment effectively blocked regional and distant metastatic dissemination in the absence of systemic toxicity in the metastatic HNSCC mouse model. The repeated administration of T22-DITOX-H6 clearly abrogates tumor invasiveness and metastatic dissemination without inducing any off-target toxicity. Thus, T22-DITOX-H6 holds great promise for the treatment of CXCR4^+^ HNSCC patients presenting worse prognosis.

## 1. Introduction

Head and neck squamous cell carcinoma represents a major cause of mortality, with more than 850,000 new cases and 400,000 deaths worldwide in 2020 [1]. Current treatment, consisting of a combination of surgery, chemotherapy, and radiotherapy, achieves loco-regional control of the disease in a variable proportion of patients. Treatment regimens for recurrent and metastatic HNSCC patients present low response rates and limited survival benefits. Remarkably, around 60% of HNSCC patients develop loco-regional recurrence and up to 30% develop distant metastases after treatment, representing the leading cause of patient mortality [2].

Different molecular pathways have been related to metastatic dissemination in HNSCC, including the transforming growth factor β (TGF-β), the fibroblast growth factor receptor (FGFR), and the chemokine receptor 4 (CXCR4), among others [3,4,5,6,7]. CXCR4 and its ligand, CXCL12, play a key role in the carcinogenic process. CXCR4 is also overexpressed in more than 20 cancer types compared to the normal organs, including HNSCC [8]. Importantly, our group and others have previously reported that CXCR4 overexpression in HNSCC primary tumors correlates with loco-regional and distant recurrence and has an impact on patient prognosis [9,10]. Moreover, CXCR4 overexpression has also been related to higher tumor grade, lymph node metastasis, and poor overall survival [11]. Thus, in recent decades, CXCR4 has been exploited as a molecular target for cancer treatment. Research has focused on the development of CXCR4 antagonists, mainly small molecules, peptides, and antibodies, that can act directly on tumor cells or by regulating the tumor microenvironment [8,12]. Currently, plerixafor (AMD3100) remains the only CXCR4 antagonist on the market. Many other inhibitors have been designed with enhanced properties. Among them, polymeric plerixafor (PAMD) represents a promising strategy, presenting an improved toxicity profile and an enhanced anti-metastatic effect [13,14]. However, most of these inhibitors still present low tolerability and short half-life in circulation. Furthermore, current clinical trials involving CXCR4 antagonists are used in combination with conventional chemotherapeutic drugs.

Targeted drug delivery to CXCR4 represents a promising alternative to molecularly targeted therapy via CXCR4 inhibitors. While the latter only focuses on inhibiting CXCR4 signaling, our approach is to deliver cytotoxic compounds directly to CXCR4-overexpressing cells, aiming to selectively deplete these cancer cells, which display stem-cell-like properties and enhanced metastatic potential [11,15,16]. In addition, targeted drug delivery theoretically enables the use of higher doses while reducing off-target effects and toxicity, which are major problems of current chemotherapy [17,18]. In this framework, our group designed the T22-DITOX-H6 nanotoxin, which incorporates the T22 peptide, a CXCR4 ligand, to selectively target CXCR4-overexpressing cells, fused to the catalytic domain of the diphtheria toxin. This platform aims to deliver the cytotoxic compound selectively to CXCR4-overexpressing cancer cells without off-target toxicity in non-tumor bearing organs. This study investigates the potential use of T22-DITOX-H6 nanotoxin to prevent regional and distant metastasis in a HNSCC mouse model in the absence of systemic toxicity. To our knowledge, this is the first study involving CXCR4-targeted protein nanoparticles for the treatment of HNSCC metastatic development, which holds great promise as a future therapy for HNSCC patients.

## 2. Materials and Methods

### 2.1. Production, Purification, and Characterization of Nanoparticles

T22-DITOX-H6 protein nanoparticles were recombinantly produced in *Escherichia coli*, purified, and characterized as previously described [19]. T22-DITOX-H6 nanotoxin monomers self-assemble into 38- and 90-nanometer nanoparticles [19].

### 2.2. Cell Lines and Culture

UM-SCC-74B (74B) human-papillomavirus-negative (HPV^−^) HNSCC cell line [20] was kindly provided by Dr. Gregory Oakley. The 74B-Luci cell line was obtained by lentiviral transduction with the plasmid pLenti-III-UbC-luc (abm, Vancouver, BC, Canada) as already described in previous work [21]. The 74B-Luci cell line was cultured in Dulbecco’s Modified Eagle’s Medium (DMEM) (Gibco, Thermo Fisher Scientific, Waltham, MA, USA) supplemented with 10% fetal bovine serum (FBS), 100 U/mL penicillin/streptomycin, and 2 mM glutamine (Gibco, Thermo Fisher Scientific, Waltham, MA, USA) and incubated at 37 °C and 5% CO_2_ in a humidified atmosphere. CXCR4 expression in 74B-Luci has already been evaluated in previous work [21].

### 2.3. In Vivo Experiments

For all experiments, four-week-old female mice weighing 18–25 g were purchased from Charles River (Saint Germain-Nuelles, France). Animals were housed in a specific pathogen-free (SPF) environment with sterile food and water ad libitum. All animal experiments were approved by the Hospital de la Santa Creu i Sant Pau Animal Ethics Committee (Ethical approval code 9721, 20 February 2018). Animal body weight was evaluated throughout the course of the experiments to ensure animal welfare. A 10% weight loss was considered the humane endpoint for the experiments. For the study of the CXCR4 expression in the invasive fronts of the tumors, 3 × 10^5^ 74B-Luci cells were ortothopically inoculated (tongue) in NSG mice (NOD-scid IL2Rgamma^null^) (*n* = 3). Primary tumors were collected for later analyses 7 days after the cell inoculation, when animals started losing weight as a consequence of primary tumor growth.

To evaluate the effect of T22-DITOX-H6 repeated administration in the invasive fronts of the primary tumors, Swiss nude mice (NU(Ico)-*Foxn1^nu^*) (*n* = 8) were orthotopically inoculated with 1 million 74B-Luci cells. One day after the implantation, animals were randomized into two groups (*n* = 4) and intravenously administered up to five doses of either buffer (166 mM NaCO_3_H, pH 8) or 10 µg of T22-DITOX-H6 every day. Seven days after the tumor implantation, animals were euthanized, and organs were collected for histopathological analyses.

The antimetastatic effect of T22-DITOX-H6 was assessed in a disseminated mouse model that replicates the metastatic pattern of the disease. To this end, NSG mice (NOD-scid IL2Rgamma^null^) (*n* = 14) were inoculated with 5 × 10^4^ 74B-Luci cells in the tongue. Forty-eight hours after tumor implantation, animals were randomized into control and treated groups (*n* = 7). Control animals were intravenously administered buffer (166 mM NaCO_3_H, pH 8) and treated animals 10 µg of T22-DITOX-H6 every other day for up to 14 doses. Metastatic dissemination to the cervical lymph nodes was semi-quantitatively evaluated every week (days 2, 8, 16, 22, and 30 post-tumor implantation) by measuring tumor cells’ bioluminescent signal (BLI, total radiance photons in the region of interest (ROI)) using the IVIS^®^ Spectrum 200 (PerkinElmer, Waltham, MA, USA). For that, mice were intraperitoneally injected with firefly D-luciferin (2.25 mg/mouse, PerkinElmer) 5 min before IVIS evaluation and anesthetized with 3% isoflurane. Thirty days after the beginning of the experiment, when cervical lymph node infiltration started to cause distress in the mice, animals were euthanized and primary tumor, cervical lymph nodes, lungs, and liver BLI were semi-quantified ex vivo in the whole area of the tumor/organ. Next, primary tumors and other relevant organs were collected in 4% formaldehyde for further analysis. All BLI measurements were performed in the luminescence/photograph mode with the auto exposure setting. BLI images were analyzed with Living Image^®^ Analysis Software (PerkinElmer). Results were expressed as total flux of BLI (photons/second; radiance photons) ± SEM.

### 2.4. Histopathology, Immunofluorescence, and Immunohistochemical Analysis

Four-micrometer paraffin-embedded sections obtained from tumors and organs extracted from the animals were utilized for all histopathological and immunostaining analyses.

Colocalization of CXCR4 and Human Vimentin in the invasive front of tumor tissues was performed by immunofluorescence. Paraffin-embedded tumor sections were heated for 1 h at 60 °C, dewaxed and rehydrated. Samples were subjected to antigen retrieval using Tris-EDTA buffer, pH 9.0 (Invitrogen, Thermo Fisher Scientific, Waltham, MA, USA) in a Decloaking Chamber™ NxGen (Biocare medical, Concord, CA, USA) at 110 °C for 20 min. Blockage was performed by incubating the samples in TBS + 0.5% TritonX-100 + 3% donkey serum for 1 h at room temperature. Next, samples were incubated with the primary antibodies human vimentin mouse IgG (ready to use, Dako, Glostrup, Denmark) and CXCR4 rabbit IgG (1:250, Abcam, Cambridge, UK) overnight at 4 °C. Tissue sections were then incubated with the secondary antibodies anti-mouse IgG-Alexa Fluor^®^ 546 (1:200, Abcam) and anti-rabbit IgG-Alexa Fluor^®^ 488 (1:200, Abcam) for 2 h at room temperature. Before mounting, the tumor sections were stained with 0.5 μg/mL DAPI (Sigma-Aldrich, Sant Louis, MO, USA) for 10 min at room temperature. Samples were visualized by fluorescence microscopy and representative pictures were taken with an Olympus DP73 digital camera (Olympus Corporation, Tokyo, Japan) and analyzed using Fiji, ImageJ software (National Institutes of Health, Bethesda, MD, USA).

For histopathological analyses, organ sections were stained with hematoxylin eosin (H&E) and analyzed by two independent observers. CXCR4 (1:200, Abcam. Retrieval pH high, Dako, Glostrup, Denmark) and Human Vimentin (ready to use, Dako, Glostrup, Denmark) immunohistochemical (IHC) staining were performed in a Dako Autostainer Link48 (Glostrup, Denmark), following the manufacturer’s instructions. Representative images were captured using an Olympus DP73 digital camera and processed with Olympus CellD Imaging 3.3 software (Olympus Corporation, Tokyo, Japan).

### 2.5. Statistical Analysis

Data were represented as mean ± standard error (SEM). Statistical analyses were performed using the GraphPad Prism 5 software (GraphPad, San Diego, CA, USA). Results were analyzed by Scheirer–Ray–Hare test, Fisher’s test, and Mann–Whitney test. Differences were considered statistically significant when *p*-values < 0.05.

## 3. Results

### 3.1. CXCR4^+^ Tumor Cells Are Enriched in the Tumor Budding in a HNSCC Mouse Model

Since CXCR4 overexpression in tumor tissue has been related to enhanced migration, metastatic potential, and a higher risk of recurrence in HNSCC, we wanted to evaluate its suitability as a receptor for targeted drug delivery. Considering the relevance of the CXCR4 receptor in HNSCC prognosis, we wanted to further study CXCR4 expression in a HNSCC orthotopic mouse model. For that, human HNSCC 74B-Luci cells were inoculated orthotopically in the tongues of the mice to generate primary tumors. Both CXCR4 and human vimentin IHC staining were performed in consecutive primary tumor slides. Human vimentin IHC staining was used as a marker for the selective detection of tumor cells, since the 74B cell line constitutively expresses vimentin. Moreover, the anti-human-vimentin antibody utilized does not cross-react with mouse vimentin; thus, it is able to detect cancer cells even as single cells in mouse tissues. Remarkably, although the vast majority of the cancer cells within the primary tumor were CXCR4^−^ (Appendix A), when observing the tumor margin, several single cells and cell clusters that expressed CXCR4 invading the stromal tissue of the tumor edge were detected (Figure 1A). Moreover, some of these CXCR4^+^ cells were also positive for human-vimentin expression, implying that they were cancer cells (Figure 1A). In order to confirm these observations, CXCR4 and human vimentin co-immunofluorescent staining was performed on the tumor samples. The CXCR4^+^ vimentin^+^ cells were observed in the tumor budding in the primary tumor margin, demonstrating that the CXCR4^+^ cells previously identified by IHC were indeed human cancer cells (vimentin^+^) (Figure 1B). These results clearly suggest that CXCR4 plays an important role in the invasion and dissemination of cancer cells from primary tumor sites, which has already been demonstrated for other cancer types [11,15,16].

### 3.2. T22-DITOX-H6 Nanotoxin Treatment Abrogates Tumor-Cell Invasion In Vivo

The detection of these CXCR4^+^ tumor cells in the invasive front of the primary tumors exhorted us to investigate the potential anti-invasive effect of the T22-DITOX-H6 nanotoxin. T22-DITOX-H6 includes the CXCR4 ligand T22, fused to the cytotoxic domain of the diphtheria toxin, which is able to selectively internalize and eliminate CXCR4^+^ HNSCC cancer cells [21,22]. In this context, we generated an orthotopic HNSCC mouse model through the inoculation of the 74B-Luci cells in the mouse tongues. The animals were administered up to five doses of either buffer (166 mM NaCO_3_H, pH 8) or 10 µg of T22-DITOX-H6 on a daily basis and euthanized 48 h after the end of the treatment (day 7 after tumor cell inoculation), at which point the tumors and other relevant organs were collected for later analyses (Appendix A). Human vimentingIHC staining revealed that nanotoxin repeated administration clearly diminished the number of cancer cells (vimentin^+^) in the tumor invasive front (Figure 2A,B). In agreement with this finding, the CXCR4^+^ cells in the tumor budding were also reduced upon nanotoxin treatment (Figure 2C,D). As expected, no variation in primary tumor volume was observed between the control and nanotoxin-treated animals, as the CXCR4 expression within the tumor tissue was negligible (Appendix A). Altogether, these findings clearly suggest that successive T22-DITOX-H6 administration effectively eliminates CXCR4^+^ invasive cancer cells endowed with metastatic potential.

### 3.3. T22-DITOX-H6 Repeated Dosage Inhibits Metastatic Dissemination in a HNSCC Orthotopic Mouse Model in the Absence of Systemic Toxicity

The aforementioned potent inhibition of tumor-cell invasion prompted us to further investigate the potential antimetastatic activity of T22-DITOX-H6 in an orthotopic mouse model that replicates the metastatic pattern observed in HNSCC patients. Animals were administered up to 14 doses of either buffer (166 mM NaCO_3_H, pH 8) or 10 µg of T22-DITOX-H6 on alternate days. Tumor-cell dissemination was assessed weekly in vivo by measuring the bioluminescent signal emitted by tumor cells (BLIs). Forty-eight hours after the last dose (day 30 after tumor-cell inoculation), the animals were euthanized. BLI of the primary tumors and different relevant organs was evaluated ex vivo. Then, primary tumors and organswere collected for immunohistochemical studies (Appendix A).

Regional metastatic dissemination to the cervical lymph nodes was evaluated during the course of the experiment by a semi-quantitative measurement of the bioluminescent signal emitted by the tumor cells (Figure 3A–C). Remarkably, the buffer-treated animals presented greater cervical-lymph-node cancer-cell infiltration compared to the nanotoxin-treated mice (Figure 3A). Semi-quantification of the emitted bioluminescent signal by cervical lymph nodes during the experiment clearly showed that the T22-DITOX-H6 treatment abrogated cervical lymph node tumor infiltration (Figure 3B). In agreement, the area under the curve (AUC) for the lymph nodes’ bioluminescent signal was also significantly smaller in the nanotoxin-treated group (Figure 3C). The follow-up of the bioluminescent signal in vivo was further validated ex vivo at euthanasia, confirming the reduction in cervical-lymph-node tumor infiltration resulting from the nanotoxin treatment (Appendix A). Accordingly, the T22-DITOX-H6 treatment affected the percentage of animals with cervical-lymph-node dissemination, with a 57% reduction (71% of the control animals, versus 14% of the treated animals) (Figure 3D). Importantly, six animals from the nanotoxin-treated group were metastasis-free, whereas only two control animals presented no lymph-node tumor infiltration at the end of the experiment. The metastatic cells were also detected by human-vimentin IHC staining, revealing that the lymph nodes collected from the buffer-treated animals were vimentin^+^, further corroborating their infiltration by the tumor cells (Figure 3E). Remarkably, cervical lymph node infiltration was macroscopically detected in the control animals, while their nanotoxin-treated counterparts presented a normal appearance, with an absence of visible cancer masses (Figure 3F). In agreement, tumor-cell infiltration also affected the cervical-lymph-node area, with the cervical lymph nodes derived from the nanotoxin-treated animals being significantly smaller compared to their buffer-treated counterparts (Figure 3G). Thus, T22-DITOX-H6 repeated administration in the disseminated mouse model clearly inhibited regional cervical lymph node metastasis.

Furthermore, distant metastatic dissemination to the lungs and liver, two commonly observed metastatic sites in advanced-HNSCC patients, was also assessed at the end of the experiment (Figure 4). Ex vivo evaluation of the bioluminescent signal emitted by the lung (Appendix A) and liver (Appendix A) samples at the endpoint of the experiment showed a reduction in the tumor-cell dissemination upon T22-DITOX-H6 repeated administration. Lung metastatic foci were detected by human-vimentin IHC, as no macroscopic metastases were visible at first sight, neither in the buffer-treated animals, nor in the nanotoxin-treated animals. Human-vimentin immunostaining allowed the precise detection of the lung metastatic foci, which were not only formed by cancer cell clusters, but also by single cells (Figure 4A). Importantly, the repeated dosage of nanotoxin dramatically impaired lung metastatic dissemination, as five animals from the treated group presented no metastatic infiltration while no buffer-treated animals were metastasis-free after the treatment, representing a 70% reduction in the occurrence of lung metastases (100% in the buffer-treated animals, compared to 30% in the nanotoxin-treated group) (Figure 4B). Furthermore, the number of lung metastatic foci, both single-cell and cluster, observed in the animals from the control group was significantly higher than the number detected in the nanotoxin-treated group, further corroborating the antimetastatic effect of the T22-DITOX-H6 treatment (Figure 4C).

Similarly, human-vimentin IHC was also utilized to study the metastatic dissemination to the liver, since no macroscopic metastatic foci could be observed directly (Figure 4D). Remarkably, the T22-DITOX-H6 treatment clearly abrogated liver metastases. First, the percentage of animals displaying liver metastatic foci decreased as a consequence of the nanotoxin treatment, with a 43% reduction in the liver metastasis occurrence (100% metastatic animals in the buffer-treated group, versus 57% in the T22-DITOX-H6-treated group). Thus, three out of seven nanotoxin-treated animals were completely free of liver metastases (Figure 4E). In agreement with this finding, the number of both single and cluster metastatic foci was reduced in the T22-DITOX-H6 group compared to the buffer-treated animals, also implying that the nanotoxin treatment reduced the development of liver metastasis in this model (Figure 4F).

Finally, no systemic toxicity derived from the repeated administration of T22-DITOX-H6 was observed in the animals. The H&E staining of the livers and kidneys (the organs involved in drug metabolism and elimination) after treatment showed no histopathological alterations (Figure 5A). Spleen samples were also studied; since some leukocytes express CXCR4, the spleen constitutes a potential site for on-target toxicity. Importantly, the spleen H&E analysis also showed that the normal architecture of the organ was preserved after nanotoxin treatment (Figure 5A). In addition, no differences in body weight were detected between the control and T22-DITOX-H6-treated animals throughout the experiment, further confirming the lack of off-target toxicity of the treatment (Figure 5B).

Thus, intravenous T22-DITOX-H6 repeated administration dramatically blocked both regional and distant dissemination of the HNSCC cells in this orthotopic mouse model able to replicate the metastatic pattern observed in HNSCC patients. It is important to mention that no primary tumor shrinkage was observed after the treatment (Appendix A), as has been already mentioned for a previous experiment (Appendix A).

## 4. Discussion

In this work, we showed for the first time that targeted delivery of the diphtheria cytotoxic domain to CXCR4-overexpresing human HNSCC via the T22-DITOX-H6 nanotoxin effectively eliminates the cancer cells present in the invasive front of primary tumors, thus demonstrating a potent anti-invasive effect. In addition, the repeated administration of T22-DITOX-H6 in a HNSCC-disseminated mouse model that replicates the metastatic pattern found in patients achieved a potent antimetastatic effect, which included a dramatic blockage of regional lymph node dissemination and a potent inhibition of distant metastasis to the lungs and liver, without inducing systemic toxicity in the animals. Remarkably, the nanotoxin treatment presented no effect on the primary tumor, suggesting that T22-DITOX-H6 is capable of selectively eliminating the CXCR4-overexpressing cells responsible for the metastatic process. Importantly, metastatic dissemination still represents the main cause of HNSCC patient mortality [23,24,25], highlighting the necessity for novel therapeutic strategies, such as our targeted drug delivery approach.

Notably, we detected CXCR4^+^ human HNSCC cells in the edges of primary tumors in an orthotopic mouse model. These CXCR4^+^ cancer cells, located in the tumor front, are empowered with a greater invasive and metastatic potential, and have also been described in other cancer types, in which they have been identified as cancer stem cells (CSCs) [26,27,28]. In this framework, repeated T22-DITOX-H6 administration effectively eliminated the CXCR4^+^ HNCSS cancer cells in the invasive front of the primary tumors. Thus, the selective elimination of these highly metastatic CXCR4^+^ cancer cells would potentially block metastatic dissemination.

Consequently, T22-DITOX-H6 nanotoxin repeated intravenous administration in a HNSCC mouse model induced a potent blockage of tumor metastasis, both regional and distant, in the absence of systemic toxicity. In vivo BLI was utilized to semi-quantitatively evaluate the regional tumor dissemination to the cervical lymph nodes. Although it is an extremely useful technique for preclinical cancer studies, in vivo BLI is limited by spatial resolution and poor signal tissue penetration, which prevented us from detecting in vivo tumor cell infiltration in the organs located deeper in the mouse body, such as lungs and liver [29]. To overcome these limitations, a semi-quantitative ex vivo BLI assessment of the relevant explanted organs was performed at euthanasia, revealing an effect of the nanotoxin treatment in blocking the metastatic dissemination. Moreover, the BLI semi-quantitative measures were further corroborated by the IHC analyses, which showed that, in fact, the nanotoxin-treated animals presented a reduction in both regional and distant tumor metastasis. Importantly, the nanotoxin treatment induced a 57% reduction in the regional dissemination to the cervical lymph nodes, a major metastatic site in HNSCC patients. Over 40% of HNSCC patients present cervical lymph node dissemination at diagnosis, and up to 30% of early-stage patients are still at risk of developing regional metastases during the disease, dramatically affecting their prognosis and survival [30,31]. In addition, nanotoxin treatment also reduced the distant metastatic dissemination to both the lungs and liver, with reductions of 70% and 43%, respectively. Although distant dissemination is not especially frequent at presentation, up to 30% of HNSCC patients develop metastases in the time course of their disease, presenting a very poor prognosis with a median overall survival of less than one year [2,32]. Moreover, the vast majority of recurrent and metastatic HNSCC patients are only candidates for palliative treatment, emphasizing the urgent need for novel curative therapies [32,33]. However, it is important to comment that a percentage of the nanotoxin-treated animals still presented metastases after treatment, suggesting that targeting the CXCL12/CXCR4 axis might not be sufficient to completely ablate metastatic dissemination. These results pave the way for further exploration of the combination of T22-DITOX-H6 treatment with other therapies, such as targeted drugs against TGF-β or FGFR that are also involved in the HNSCC metastatic spread [3,4,5,6,7].

Current HNSCC treatment still mainly relies on conventional chemotherapeutic drugs, as well as molecularly targeted drugs (cetuximab) and immune checkpoint inhibitors (pembrolizumab) [32,33]. Although the incorporation of novel targeted therapies has improved patient survival, the response rates to both cetuximab and pembrolizumab are quite low [34,35]. Chemotherapeutic drugs lack selectivity, thus inducing important off-target toxicities in non-tumor-bearing organs, compromising patients life quality [36,37]. Moreover, therapy resistance, both to chemotherapy and to molecular therapies, is an important drawback of current treatments, preventing complete remission and leading to recurrence [38,39]. In this framework, the T22-DITOX-H6 nanotoxin represents a promising approach for HNSCC treatment, as it aims to deliver cytotoxic compounds exclusively to CXCR4^+^ cancer cells. Our previous work demonstrated the selective accumulation of nanoparticles in CXCR4-overexpressing tumor tissues [21], together with a CXCR4-dependent cytotoxic effect and a potent antitumor effect in vivo [22]. Here, we demonstrate, for the first time, that the T22-DITOX-H6 nanotoxin induces potent anti-invasive and antimetastatic effects in vivo. Other targeted drug delivery strategies have also been explored for HNSCC treatment. Different immunotoxins, such as VB4-845 and SS1P, both including the *Pseudomonas aeruginosa* exotoxin fused to anti-EpCAM or mesothelin targeting moieties, have undergone clinical trials for the treatment of HNSCC. However, none of them has yet reached the market, due to immunogenicity, off-target toxicity concerns, and a lack of antitumor effect due to poor tumor uptake [40,41]. By contrast, our T22-based nanotoxin presents interesting features, including efficient single-step production and purification in recombinant bacteria, easy production scale-up, biocompatibility, and a lack of off-target toxicity [19]. Moreover, while immunotoxins display only one targeting moiety per molecule and a low cytotoxic payload [42,43], this T22-based nanotoxin is produced by the self-assembly of multiple monomers, conferring superselectivity derived from the display of multiple T22 ligands [44]. However, it is relevant to take into consideration that the antineoplasic effect of T22-DITOX-H6 has only been evaluated in immunodeficient mice displaying a compromised immune system. Since immunogenicity represents a major drawback of immunotoxins, a thorough evaluation of T22-DITOX-H6’s effect on the immune system is key to its further clinical translation. In addition, different immune cells, such as lymphocytes, constitutively express CXCR4; they thus represent potential targets for nanotoxin treatment. To assess these questions, our group is currently developing syngeneic mouse models for different cancer types, including HNSCC, to study the effect of nanotoxin treatment on immunocompetent animals. Nonetheless, the potent anti-invasive and anti-metastatic effect demonstrated in the present article clearly supports the relevance of T22-DITOX-H6 as a promising treatment for HNSCC patients. Altogether, T22-DITOX-H6 holds great promise for future clinical translation.

## 5. Conclusions

In conclusion, CXCR4 expression in the invasive front of HNSCC primary tumors supports the previously reported implication of the receptor in the invasive and metastatic processes. Moreover, CXCR4 overexpression in HNSCC cancer cells compared to healthy tissue makes it an ideal entryway for targeted drug delivery. Thus, T22-DITOX-H6’s ability to eliminate CXCR4^+^ cancer cells presenting a more invasive and metastatic phenotype blocks HNSCC’s invasiveness and its metastatic dissemination to the cervical lymph nodes, lungs, and liver, in the absence of histopathological alterations. Altogether, the T22-DITOX-H6 nanotoxin represents a promising alternative treatment for HNSCC patients that are still at risk of developing metastatic disease and recurrence, which significantly compromise their clinical outcome and survival.

## Figures and Tables

**Figure 1 pharmaceutics-14-00887-f001:**
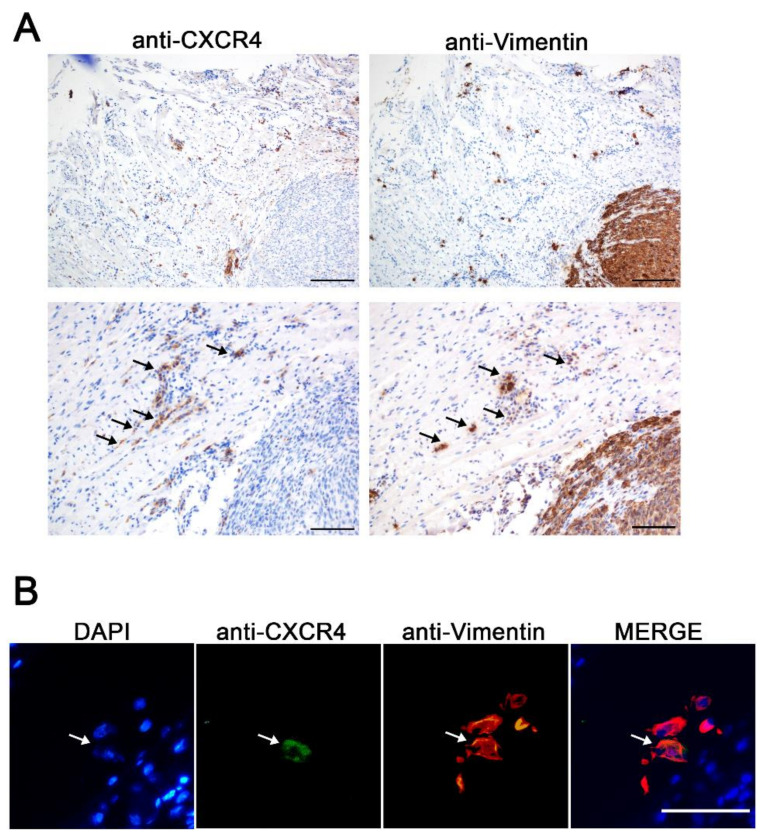
CXCR4 expression in the tumor invasive front generated in the HNSCC orthotopic mouse model. (**A**) Representative CXCR4 and human-vimentin IHC images of the tumor budding showing the presence of CXCR4^+^ cancer cells invading the surrounding tissue. Scale bars = 200 µm and 100 µm. (**B**) CXCR4 and human-vimentin immunofluorescence staining in the invasive front of the orthotopic tumor samples. Scale bars = 50 µm.

**Figure 2 pharmaceutics-14-00887-f002:**
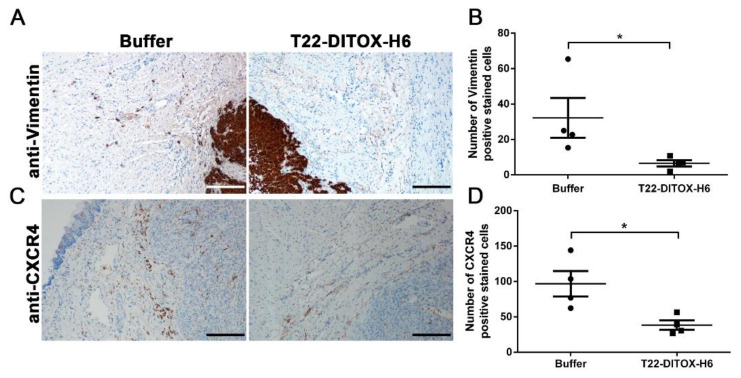
Anti-invasive T22-DITOX-H6 effect in the tumor front in a HNSCC orthotopic mouse model. (**A**) Representative human-vimentin IHC images of tumors collected from buffer- and T22-DITOX-H6-treated animals, showing a reduction in the number of human-vimentin-positive stained cells in the invasive front of the tumors after the nanotoxin treatment. Scale bar = 200 µm. (**B**) Quantification of the number of human-vimentin-positive stained cells in the tumor budding in the control and nanotoxin-treated tumors. (**C**) CXCR4 IHC analysis of the invasive front of tumors derived from control and nanotoxin-treated mice, displaying a reduction in the number of CXCR4^+^ cells upon T22-DITOX-H6 treatment. Scale bar = 200 µm. (**D**) Quantification of the number of CXCR4-positive stained cells in the aforementioned CXCR4 IHC images. * *p* < 0.05; *n* = 4 per group (total animal number 8). Statistical analysis was performed by Mann–Whitney test. Error bars indicate SEM.

**Figure 3 pharmaceutics-14-00887-f003:**
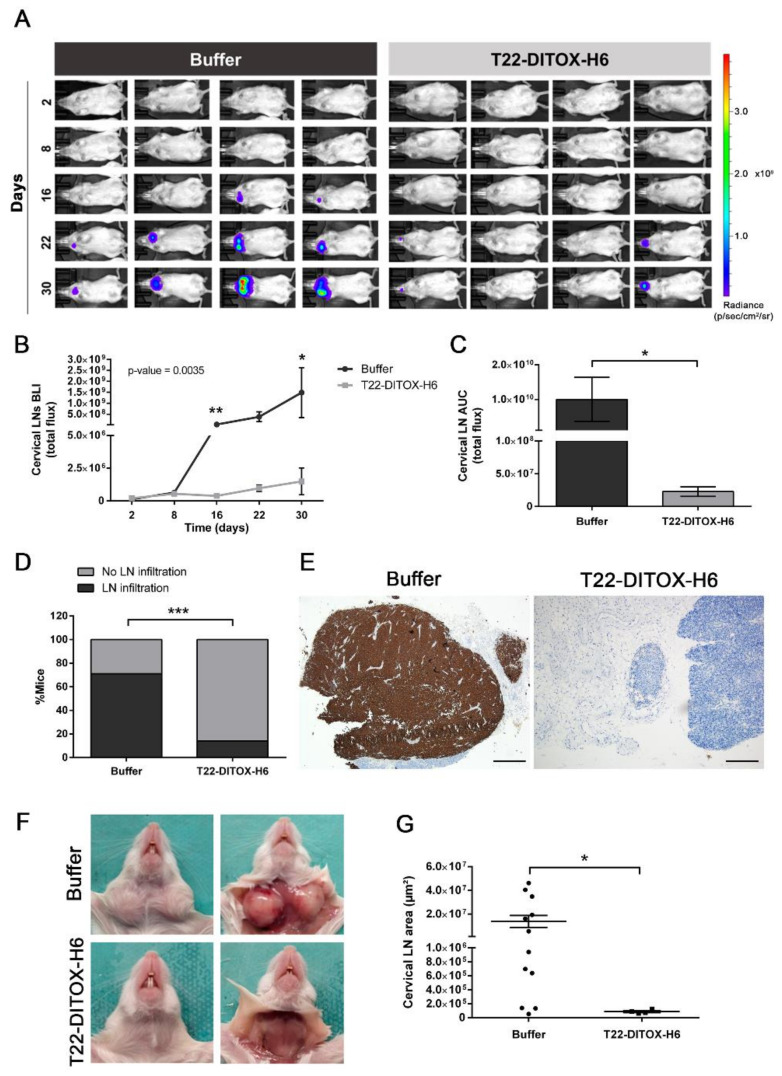
T22-DITOX-H6 repeated administration reduces the occurrence of regional dissemination to the cervical lymph nodes in a HNSCC-disseminated mouse model. (**A**) Bioluminescence intensity (BLI) emitted by 74B-Luci cancer cells during the experiment in the buffer- and T22-DITOX-H6-treated animals. (**B**) Semi-quantification of the emitted BLI in the cervical lymph nodes (LNs) throughout the experiment in the control and treated groups. (**C**) Area under the curve (AUC) of the registered BLI emitted by cervical lymph nodes (LN) in the time course of the experiment for both control and nanotoxin-treated animals. (**D**) Percentage of the animals presenting cervical-lymph-node (LN) infiltration at the endpoint of the experiment in the buffer- and T22-DITOX-H6-treated groups. (**E**) Human vimentin IHC analysis of cervical lymph node samples from control and treated animals at the endpoint of the experiment (day 30 post-tumor-cell inoculation). Scale bars = 500 µm and 200 µm. (**F**) Representative images of the cervical lymph nodes (LN) from a buffer-treated animal (up) and a nanotoxin-treated animal (down) at euthanasia. Animals from the buffer-treated group presented macroscopic infiltrated lymph nodes. (**G**) Quantification of the area of the cervical lymph nodes observed in the human-vimentin IHC samples collected from buffer- and T22-DITOX-H6 groups. * *p* < 0.05; ** *p* < 0.01; *** *p* < 0.001; *n* = 7 per group (total animal number 14). Statistical analysis was performed by Scheirer–Ray–Hare test, Mann–Whitney test, and Fisher’s test. Error bars indicate SEM.

**Figure 4 pharmaceutics-14-00887-f004:**
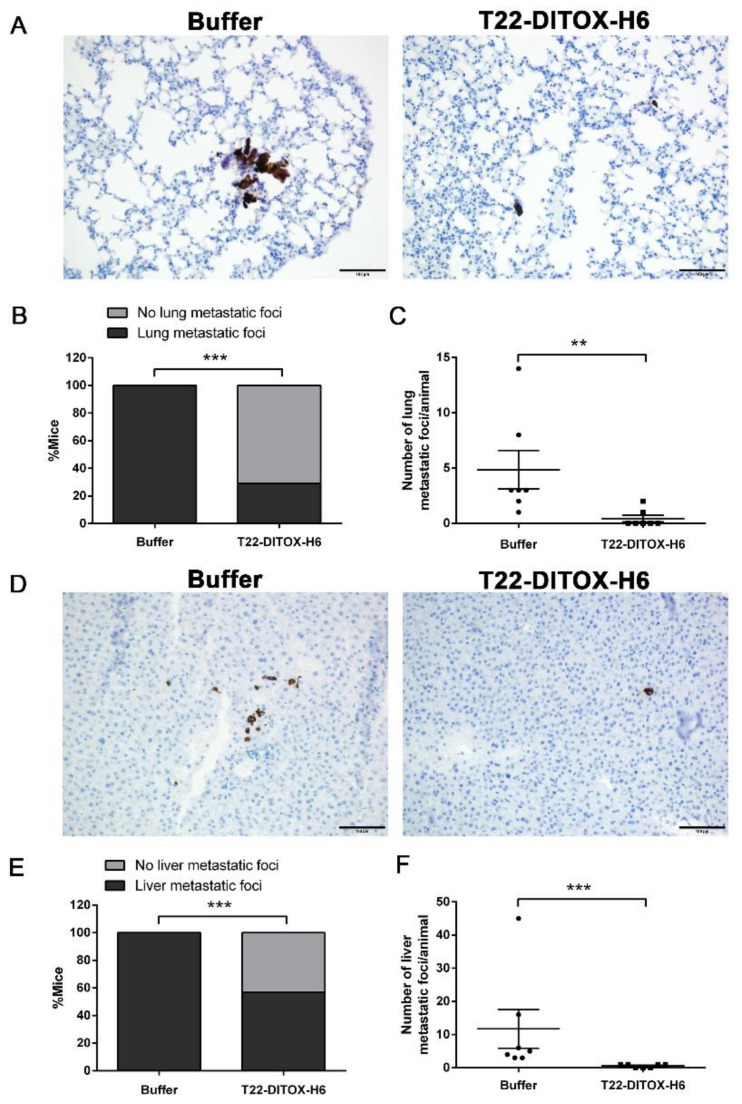
T22-DITOX-H6 repeated administration inhibits distant metastatic dissemination to lungs and liver in a HNSCC-disseminated mouse model. (**A**) Representative human-vimentin IHC images of lung metastatic foci in samples obtained from buffer- and nanotoxin-treated mice. Scale bars = 100 µm. (**B**) Percentage of the animals from control and treated groups displaying lung metastases detected by human-vimentin IHC. (**C**) Quantification of the number of lung metastatic foci in each animal from the buffer- and T22-DITOX-H6-treated groups. (**D**) Human-vimentin IHC images showing the metastatic foci in the liver samples collected from control and treated mice. Scale bars = 100 µm. (**E**) Percentage of the animals from buffer- and T22-DITOX-H6 groups presenting liver metastases detected by human-vimentin IHC. (**F**) Quantification of the number of liver metastatic foci in each animal from the buffer- and nanotoxin-treated groups. ** *p* < 0.01; *** *p* < 0.001; *n* = 7 per group (total animal number 14). Statistical analysis was performed by Mann–Whitney test and Fisher’s test. Error bars indicate SEM.

**Figure 5 pharmaceutics-14-00887-f005:**
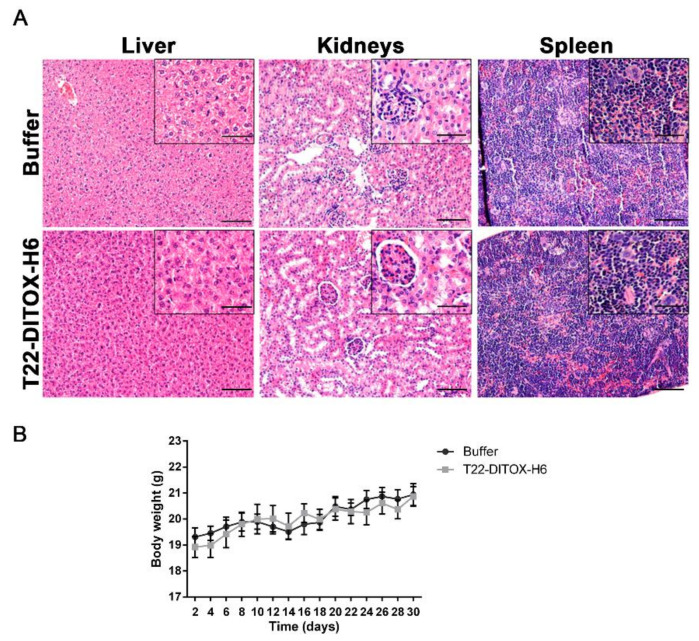
Evaluation of the systemic toxicity derived from T22-DITOX-H6 administration in the HNSCC-disseminated mouse model. (**A**) Histopathological analysis by H&E staining in liver, kidneys, and spleen samples collected from buffer- and T22-DITOX-H6-treated groups. Scale bars = 100 µm and 50 µm (zoom in) (**B**) Body weights of buffer- and nanotoxin-treated animals over the course of the experiment. Error bars indicate SEM.

## Data Availability

Data is available from the corresponding author upon justified demand.

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
