# Peer review of "A Novel CXCR4-Targeted Diphtheria Toxin Nanoparticle Inhibits Invasion and Metastatic Dissemination in a Head and Neck Squamous Cell Carcinoma Mouse Model"

_pharmaceutics, 2022, doi:10.3390/pharmaceutics14040887_

Round 1

Reviewer 1 Report

This is a very interesting study from Rioja-Blanco et al.. Here are my comments for major revision :

1) p.3 l. 108: parameters of BLI acquisition should be described. the days the BBLI acquisitions were performed should be described in this section. Also, there are absolutely no details about the way the BLI signal was "semi-quantified": regions/volumes of interests? imaging data treatment software?

2) The paper lacks a figure introducing the experimental paradigm with animal groups and the number of animals, experiments performed, timeline, etc. which would help the reader better understand at a glance the experimental design of the study.

3) p.4 l. 157 and p.7 Fig. 3B and C: it is highly questionable the authors chose a Mann-Whitney test given the experimental design. A 2-way ANOVA or a mixed-model analysis followed by a post-hoc multiple comparison test would sound a lot more appropriate to evaluate the effect along time and experimental groups.

4) p.3 l. 115: animal body weight was followed but not exploited as a result in the result section. Please clarify the purpose of following the body weight.

5) p.7 Fig.3D: this presentation doesn't allow showing error bars. Please chose another presentation to enable a more scientific graphical presentation of the results.

6) p.7 Fig. 3E: the figure legend should precise the day(s) the experiments were performed.

7) p.7 fig. 3B legend and discussion: BLI does not allow absolute quantification in vivo at all, mainly due to a major attenuation of tissues with this type of imaging modality. The authors should clearly discuss this limitation in the discussion and choose a more appropriate term such as semi-quantification when writing about their methods and results. At the final point, for example, it would have been very interesting to image explanted organs with BLI to increase sensitivity.

8) Discussion: the authors did not discuss their study limitations at all. This point is critical.

Author Response

Response to Reviewer 1 Comments

This is a very interesting study from Rioja-Blanco et al.. Here are my comments for major revision:

We thank the reviewer for the careful and insightful review of our work and the provided suggestions that have helped us to enhance the manuscript's quality. Following his/her recommendations, we have improved the introduction and discussion sections of the manuscript, as well as added 2 new supplementary figures to further support the results and conclusions. Lastly, English language and style have been checked and edited.

Point 1: p.3 l. 108: parameters of BLI acquisition should be described. the days the BBLI acquisitions were performed should be described in this section. Also, there are absolutely no details about the way the BLI signal was "semi-quantified": regions/volumes of interests? imaging data treatment software?

First of all, we would like to apologize for the lack of detail regarding BLI acquisition. We have now thoroughly described the BLI imaging assessment in the materials and methods section of the manuscript. In summary, the evolution of the cervical lymph node dissemination was evaluated in vivo on a weekly basis (days 2, 8, 16, 22, and 30 post-tumor implantation). All images were analyzed at the same time. For that, we selected a ROI (region of interest) that comprised the cervical lymph node area, the same ROI was selected in all images to ensure that the area was the same and thus could be compared between images. We represented the results as Total flux of BLI (photons/second; Radiance photons). In the case of the ex vivo analysis at the end of the experiment (day 30), after euthanasia, we explanted the relevant organs and measured their emitted BLI. In a similar way to the in vivo follow up, BLI was semi-quantified selecting the same ROI in all the images. Again, results were represented as Total flux (photons/second). In all cases, images were analyzed using the Living Image® Analysis Software from PerkinElmer.

We also want to clarify that we are aware that BLI data is merely semi-quantitative and an indirect measure of the presence of tumor cells in the different organs. For that reason, we always use immunohistochemistry as an independent technique to evaluate the metastatic dissemination in the animal model. We have now clarified that BLI measurements are semi-quantitative in the new version of the manuscript.

Point 2: The paper lacks a figure introducing the experimental paradigm with animal groups and the number of animals, experiments performed, timeline, etc. which would help the reader better understand at a glance the experimental design of the study.

A new supplementary figure (Figure S2) has now been introduced explaining the experimental design of each experiment, including the timeline, treatment administration, and BLI acquisition. We hope that this new figure helps to gain a better understanding of our study.

Point 3: p.4 l. 157 and p.7 Fig. 3B and C: it is highly questionable the authors chose a Mann-Whitney test given the experimental design. A 2-way ANOVA or a mixed-model analysis followed by a post-hoc multiple comparison test would sound a lot more appropriate to evaluate the effect along time and experimental groups.

Certainly, we have not considered time as a second variable, since we were only evaluating the effect of the treatment at each given time in the two experimental groups, buffer vs. T22-DITOX-H6. Since the population does not follow a normal distribution, we have performed a non-parametric Mann-Whitney test comparing both groups at each time. However, we agree with the reviewer in pointing out that we must consider the influence of the time, as well as the possible interaction of the two variables in the results.

Following his/her recommendations, we have now performed a Scheirer-Ray-Hare test, the non-parametric equivalent of the two-way ANOVA, to evaluate the effect of the treatment over time. The results indicate that there is a significant effect of both time (p-value < 0.0001) and treatment (p-value = 0.0035), as well as their interaction (p-value = 0.0076) in the BLI evolution. To compare the effect of the treatment (buffer vs. T22-DITOX-H6) in the lymph node BLI, we performed a post-hoc multiple comparison test using the Sidak method. This test revealed the existence of statistical significant differences, attributable to the treatment, between the control and the nanotoxin-treated groups at the end of the experiment (16 and 30 days), as it was previously predicted. We have now included this new analysis in the revised version of the manuscript.

In case of figure 3C, since we are only comparing the area under the curve (AUC) for the two groups, we only have one variable (treatment), thus we believe that the Mann-Whitney test represents the correct statistical test for a non-Gaussian population. 

Point 4: p.3 l. 115: animal body weight was followed but not exploited as a result in the result section. Please clarify the purpose of following the body weight.

We apologize for the lack of clarity regarding the evaluation of the animal body weight in the different experiments. In all the experiments involving animal models, variation on the body weight was followed to ensure animal welfare, since weight loss represents a humane endpoint criteria. In the case of the experiment aimed at evaluating the anti-metastatic effect of the nanotoxin, since the treatment was intensive, we wanted to ensure that the nanotoxin repeated administration did not present any toxic effect in the animals. For that reason, we have included the evolution of the body weight of the animals in Figure 5B, not observing any differences between control and T22-DITOX-H6 treated animals. The lack of systemic toxicity was further validated by histopathological analysis in HE stained relevant organs (Figure 5A). We have now modified the materials and methods section of the manuscript to clarify the purpose of assessing the animal body weight.

Point 5: p.7 Fig.3D: this presentation doesn't allow showing error bars. Please chose another presentation to enable a more scientific graphical presentation of the results.

Figure 3D represents the percentage of animals in each experimental group (control vs. T22-DITOX-H6) that presented lymph node metastasis at the end of the experiment. Since this is a categorical variable (metastasis or no metastasis detected), we are unable to represent this result in any other way that allows us to show error bars (no mean can be calculated). Additional ways to evaluate the lymph node dissemination are the semi-quantitative BLI data and the measurement of the cervical lymph nodes area that have already been incorporated in Figure 3, both showing a clear effect of the nanoparticle treatment.

Point 6: p.7 Fig. 3E: the figure legend should precise the day(s) the experiments were performed.

All immunohistochemical evaluations are performed at the endpoint of the experiments, in this case at day 30 post-tumor implantation, when the animals were euthanized and the relevant organs fixed and paraffin embedded for later analysis. We have now included this information in the figure legend.

Point 7: p.7 fig. 3B legend and discussion: BLI does not allow absolute quantification in vivo at all, mainly due to a major attenuation of tissues with this type of imaging modality. The authors should clearly discuss this limitation in the discussion and choose a more appropriate term such as semi-quantification when writing about their methods and results. At the final point, for example, it would have been very interesting to image explanted organs with BLI to increase sensitivity.

As stated previously, we are aware of the limitations of measuring the BLI in vivo. Indeed, we could only detect in vivo the BLI signal from the cervical lymph nodes, but not from other organs like lungs and liver, since they are located deeper in the mouse body, mainly due to BLI´s poor signal penetration. In fact, the term “semi-quantify” certainly describes better the limitations of such measurements and we have incorporated it to the manuscript as suggested. Moreover, we have now also included a new supplementary figure (Figure S4) displaying the ex vivo evaluation of the BLI emitted by the organs at the endpoint of the experiment. These results further confirm the anti-metastatic effect of T22-DITOX-H6 repeated administration. Lastly, the limitations of the in vivo BLI assessment are also discussed in the new version of the manuscript.

Point 8: Discussion: the authors did not discuss their study limitations at all. This point is critical.

We would like to apologize to the reviewer for not discussing the constraints of the present study. Besides the already discussed limitations of in vivo BLI measurements, we consider that the main limitation of this work is the use of immunodeficient mouse models, that are extremely useful to study the antineoplasic effect of a certain treatment in xenografted human cancer, but are limited to evaluate its potential interaction with the immune system. For that reason, our group is currently developing syngeneic mouse models to overcome these limitations. In addition, the T22-DITOX-H6 treatment alone, although displaying a potent anti-metastatic effect, is unable to completely block metastatic dissemination in all treated animals. This fact suggests that the CXL12/CXCR4 axis is not the only pathway involved in the metastatic spread in this HNSCC mouse model. Thus, combination treatment with other therapies that target different molecular pathways involved in the development of metastasis (TGF-β, FGFR, etc.) might represent a promising approach to be investigated in the future. All these points are now discussed in depth in the discussion section of the manuscript´s new version.

Reviewer 2 Report

The manuscript descibes  the develop of a previously published protein nanoparticle, T22-DITOX-H6 (see reference 12), aiming to selectively deliver the diphtheria toxin cytotoxic domain to CXCR4+ HNSCC cells. T22-DITOX-H6 antimetastatic effect was evaluated in vivo in an orthotopic mouse model. IVIS imaging system was utilized to assess the metastatic dissemination in the mouse model. Intravenous T22-DITOX-H6 repeated administration dramatically blocks both regional and distant dissemination of the HNSCC cells in this orthotopic mouse model able to replicate the metastatic pattern observed in HNSCC patients. The manuscript is interesting, but it is only an extension of the previous published manuscript (reference 12) related to a colorectal cancer model on a HNSCC cancer cell model using the same protein nanoparticle, T22-DITOX-H6.

Author Response

Response to Reviewer 2 Comments

Comments and Suggestions for Authors

The manuscript describes the develop of a previously published protein nanoparticle, T22-DITOX-H6 (see reference 12), aiming to selectively deliver the diphtheria toxin cytotoxic domain to CXCR4+ HNSCC cells. T22-DITOX-H6 antimetastatic effect was evaluated in vivo in an orthotopic mouse model. IVIS imaging system was utilized to assess the metastatic dissemination in the mouse model. Intravenous T22-DITOX-H6 repeated administration dramatically blocks both regional and distant dissemination of the HNSCC cells in this orthotopic mouse model able to replicate the metastatic pattern observed in HNSCC patients. The manuscript is interesting, but it is only an extension of the previous published manuscript (reference 12) related to a colorectal cancer model on a HNSCC cancer cell model using the same protein nanoparticle, T22-DITOX-H6.

We appreciate the overall positive evaluation of our work by the reviewer. Although we respect the opinion of the reviewer, we consider that the present study has sufficient novelty to be considered as an independent research work regarding T22-DITOX-H6 therapeutic potential and differ from the data published by Sánchez-García, L. et al. (former reference 12, reference 19 in the manuscript´s new version). Sánchez-García, L. et al. thoroughly described the production and characterization both T22-PE24-H6 and T22-DITOX-H6 nanotoxins. However, the study merely evaluated the antitumor effect of both nanoparticles in subcutaneous colorectal cancer tumors that overexpressed CXCR4. In our opinion, our study is not just repeating the same experiments in a HNSCC mouse model, since we demonstrate that T22-DITOX-H6 repeated administrations induce a potent blockade of metastatic dissemination in a HNSCC mouse model. Moreover, for the first time we describe that even though the percent of cancer cells displaying CXCR4 expression is quite low in the primary tumor, the nanotoxin treatment effectively eliminates the CXCR4+ invasive cells present in the primary tumor front, firmly suggesting their implication in the metastatic spread. Thus, we disagree to consider this work as an extension of a previous article.

In addition, we have improved the introduction and materials and methods sections, incorporated two new supplementary figures that further support our results, and performed and in-depth discussion of the study limitations. Lastly, English language and style have been checked and edited, as requested by the reviewer.

Reviewer 3 Report

The authors presented an study of  anti-metastatic activity of   CXCR4-targeted diphtheria toxin nanoparticles against head and neck squamous cell carcinoma, which is scientifically sound and will be interesting for readers. However, some improvements should be done before the publishing of the manuscript.

MINOR

  1. Line 63 "Up to date, plerixafor (AMD3100) remains the only CXCR4 antagonist in the market. Many other inhibitors have been designed with enhanced properties. However, most of them still present low tolerability and short half-life in circulation." It is recommended to add information on polymeric plerixafor and its improved anti-metastatic activity?
  2. It is known that head and neck squamous cell carcinoma (HNSCC) metastases are linked not only to CXC12 elevated level but also with transforming growth factorβ and fibroblast growth factor receptor signaling. Authors should modify introduction and discussion parts, and make a suggestion how to fight HNSCC metastases with anti-CXCR4 treatment only?
  3. CXCR4-SDF1 axis is normally involved into stem cells trafficking, so can this process potentially be affected by T22-DITOX-H6 treatment? Why the authors did not take bone marrow samples to study potential off-target activity?
  4. Line 87 T22-DITOX-H6 nanotoxin monomers self-assemble into 38 and 90 nm nanoparticles.Reference should be included.
  5. All the manufacturers should be written properly e.g. (Company name, City, Country).

Author Response

Response to Reviewer 3 Comments

Comments and Suggestions for Authors

The authors presented a study of anti-metastatic activity of   CXCR4-targeted diphtheria toxin nanoparticles against head and neck squamous cell carcinoma, which is scientifically sound and will be interesting for readers. However, some improvements should be done before the publishing of the manuscript.

We thank the reviewer for the positive comments and the suggested corrections that have notoriously enhanced the quality of our work. We would like to mention that an improved and more in-depth introduction is now included in the manuscript´s new version, as requested by the reviewer.

MINOR

Point 1: Line 63 "Up to date, plerixafor (AMD3100) remains the only CXCR4 antagonist in the market. Many other inhibitors have been designed with enhanced properties. However, most of them still present low tolerability and short half-life in circulation." It is recommended to add information on polymeric plerixafor and its improved anti-metastatic activity?

Polymeric plerixafor (PMAD) represents a very promising alternative to conventional plerixafor, preserving the ability to inhibit CXCR4 while enhancing its anti-metastatic effect. We would like to thank the reviewer for raising this point, the potential of PAMD is truly remarkable. Please, find this information included in the manuscript´s introduction, referencing the interesting studies that first described PAMD (references 13 and 14).

Point 2: It is known that head and neck squamous cell carcinoma (HNSCC) metastases are linked not only to CXC12 elevated level but also with transforming growth factor‑β and fibroblast growth factor receptor signaling. Authors should modify introduction and discussion parts, and make a suggestion how to fight HNSCC metastases with anti-CXCR4 treatment only?

We appreciate that the reviewer raised this interesting question. The involvement of other molecular pathways apart from the CXCR4/CXCL12 axis, such as the transforming growth factor‑β and fibroblast growth factor receptor signaling among others, has been thoroughly studied in HNSCC as well as in other cancer types. We have now included this information in the introduction of the manuscript to clarify that CXCR4 is not the only molecular pathway involved in HNSCC metastatic dissemination. Thus, targeting a single signaling pathway might not be enough to completely prevent the metastatic spread. Indeed, although we report a potent anti-metastatic effect of T22-DITOX-H6, a significant percentage of the nanotoxin-treated mice still developed metastases, suggesting the mediation of different molecular pathways. This fact is not surprising, since the majority of single-agent therapies present a limited antitumor effect, forcing the administration of a combination of different therapeutic agents. In fact, a vast majority of current clinical trials are focusing on the combination of antitumor drugs with different mechanisms of action to enhance therapeutic response. These results pave the way for the future study of treatments that combine T22-DITOX-H6 and other targeted drugs, such as TGF- β or FGFR inhibitors. This information can now be found in the discussion section of the manuscript.

Point 3: CXCR4-SDF1 axis is normally involved into stem cells trafficking, so can this process potentially be affected by T22-DITOX-H6 treatment? Why the authors did not take bone marrow samples to study potential off-target activity?

We thank the reviewer for stating this very pertinent concern. As you mention, the CXCR4/CXCL12 axis plays a physiological role, for instance in stem cell trafficking and immune cell responses. For that reason, we have previously performed an exhaustive study evaluating the potential off-target toxicity of the nanotoxin in the HNSCC mouse model (Rioja-Blanco et al., J Exp Clin Cancer Res. 2022 doi: 10.1186/s13046-022-02267-8). We did not observe any alteration in bone marrow samples, neither in the different cell blood populations upon nanotoxin treatment. In addition, no on-target toxicity in bone marrow nor off-target toxicity in other relevant organs has been reported in acute myeloid leukemia (AML) cancer models upon T22-DITOX-H6 repeated administration (Pallarès V. et al., Journal of Controlled Release 2021 doi: 10.1016/j.jconrel.2021.05.014). Nonetheless, we are aware of the limitations of our immunodeficient mouse models to fully evaluate the potential on-target effect of our nanotoxins in CXCR4+ immune cells. For that, our group is currently developing syngeneic mouse models to further study the interaction of the nanotoxin with the immune system. Please, find a detailed explanation of this limitation in the discussion section of the manuscript.

Point 4: Line 87 T22-DITOX-H6 nanotoxin monomers self-assemble into 38 and 90 nm nanoparticles. Reference should be included.

We have now included the reference as requested.

Point 5: All the manufacturers should be written properly e.g. (Company name, City, Country).

Please, find all detail of the manufacturers in the materials and methods section of the manuscript´s new version.